

# Increased surface water evaporation loss induced by reservoir development on the Loess Plateau

**Yao Liu[1,2], Xianhong Xie [1,2,*], Yibing Wang[1,2], Arken Tursun[1,2], Dawei Peng[1,2], Xinran Wu[1,2]**

1. State Key Laboratory of Remote Sensing Science, Faculty of Geographical Science, Beijing Normal University, Beijing 100875, China

2. Beijing Engineering Research Center for Global Land Remote Sensing Products, Faculty of Geographical Science, Beijing Normal University, Beijing 100875, China

*Corresponding author:

Xianhong Xie (Beijing Normal University, xianhong@bnu.edu.cn)



## 1 **Abstract**

Global-scale reservoir construction has significantly enhanced local water supply
for local production and livelihoods, yet the evaporation losses from these surface water
bodies remain poorly understood, particularly in the context of climate change. The
majority of existing studies have predominantly focused on terrestrial evaporation,
overlooking the intricate evaporation dynamics within these aquatic systems. This
study addresses this gap by investigating water body evaporation in the Loess Plateau
of China, a region characterized by extensive reservoir development over the past
decades. By employing a modified Penman equation and utilizing long-term remote
sensing water body data to calculate water depths while accounting for the thermal
storage capacity of water bodies, we estimated water evaporation rates and total
evaporation volumes for the period 2000-2018. Validation against pan evaporation
observations demonstrates the efficacy of our improved approach in capturing the
evaporation patterns of diverse water bodies in the Loess Plateau. Results reveal a
subtle decreasing trend in evaporation rates across the region. However, the total
evaporation volume amounts to a substantial $4.16 \times 10^6$ m$^3$/d, with a notable upward
trend at a rate of $0.117 \times 10^6$ m$^3$/d/yr. Attribution analysis shows that while the combined
effects of climate change marginally reduced evaporation rates, the expansion of water
bodies has counteracted this trend, resulting in a significant increase in total evaporation
losses. Particularly, the development of small- and medium-sized reservoirs and check
dams is the primary driver of increased evaporation losses on the Loess Plateau. Given
comparable evaporation losses to surface water withdrawals in this region, future water





management and hydraulic projects must consider such substantial losses. This study
fills gaps in evaporation dynamics and underscores the need for integrated strategies
addressing climate change, reservoir expansion, and evaporation.
**Keywords:**
Surface water area; Water depth; Evaporation estimation; Evaporation volume;
Reservoir development; Remote sensing;



## 1. Introduction

Land surface water bodies, such as reservoirs and lakes, serve as vital sources of 'blue water' that sustain human livelihoods and production, while their evaporation processes exert significant influence on climate regulation and land surface energy partitioning across the land surface (Guan and Mascaro, 2023). A large number of studies have been predominantly focused on evaporation from vegetation and soil profile (Jian et al., 2015; Liu et al., 2024; Peng et al., 2024), with scant attention given to the evaporation loss from surface water bodies. Globally, reservoir storage increased rapidly at a rate of 27.82 $km^3$ per year from 1999 to 2018, driven by the construction of thousands of new reservoirs to address rising demands for water supply, irrigation, and energy (Li et al., 2023). However, it has been reported that large reservoirs globally evaporated about 340 $km^3/yr$ from 1985 to 2016, which is over 70% of the amount of municipal water withdrawal in 2010 (Tian et al., 2022). Moreover, the long-term average evaporation volume from lakes worldwide is estimated to be approximately $1500 \pm 150$ $km^3/yr$ (Zhao et al., 2022). Therefore, the impact of water body evaporation losses on human society should not be underestimated.

Surface/open water evaporation is influenced by a range of meteorological factors and water surface conditions. These include near-surface air temperature, relative humidity, solar shortwave radiation, and the temperature profile within water bodies (Milly and Dunne, 2020; Vystavna et al., 2021). Variations in these factors elicit corresponding changes in both the evaporation rate and evaporation volume. For



instance, an elevation in near-surface air temperature over the lakes of the Yunnan-
Guizhou Plateau has led to accelerated evaporation rates (Yang et al., 2019). Beyond
these meteorological factors, changes in surface water area also play a crucial role in
determining evaporation losses. As an illustration, the total evaporation volume from
reservoirs in China has risen, with 96% of this increase attributed to newly constructed
reservoirs (Tian et al., 2021). Similarly, in the United States, while rising temperatures
have contributed to an increase in total evaporation loss from reservoirs, this effect has
been largely counteracted by a decrease in surface area (Zhao and Gao, 2019). Another
example is the evaporation rates at Siling Co Lake on the Tibetan Plateau, which have
decreased partially due to lower wind speeds, contributing to the expansion of the lake
surface area (Guo et al., 2019).

12        In recent decades, the Loess Plateau in China has experienced significant climatic

shifts, accompanied by substantial variations in vegetation cover (Jiang et al., 2021; Wu
et al., 2020; Xie et al., 2015). A notable observation is the enhancement of land surface
evaporation, particularly attributed to increased vegetation transpiration (Jiang et al.,
2022; Peng et al., 2024). Key climate factors such as temperature, radiation, and wind
speed all play crucial roles in this process (Bai et al., 2019; Jin et al., 2017; Li et al.,
2009). However, it is imperative to recognize that water body evaporation is also a vital
component of the water cycle on the Loess Plateau, necessitating a closer examination
of its dynamics. Moreover, to mitigate soil erosion and reduce the sediment content, a
variety of large, medium and small-scale in dam and reservoir projects have been
carried out on a large scale (Fu et al., 2017). These interventions have led to notable



changes in the water landscape (Liu et al., 2023). This substantial expansion of surface
water bodies is expected to enhance water retention, thereby supporting human
livelihoods and production (Woolway et al., 2020; Zhou et al., 2020). However, it may
also exacerbate water evaporation losses, particularly under the arid climatic conditions
of the Loess Plateau, which can further accelerate surface water evaporation.
Consequently, amidst ongoing climate change and alterations in water body area, there
is a pressing need to deepen our understanding of the evolving patterns of water body
evaporation, both in terms of rate and volume.
Several methods are accessible for estimating surface water evaporation, including
the pan measurement, eddy covariance observation (EC), and hydrological model
simulations (Friedrich et al., 2018; Hollinger and Richardson, 2005; Liu et al., 2012;
Rotstayn et al., 2006; Woolway et al., 2020). Among these approaches, the hydrological
modeling stands out due to its ability to integrate various meteorological and
hydrological factors (Deng et al., 2022; Vishwakarma et al., 2022). This approach
simulates evaporation processes across diverse environmental conditions, rendering it
suitable for large-scale regions and long-term predictions. Notably, the Penman
equation is widely preferred owing to its straightforward application principles, high
degree accuracy, and broad applicability (Fuentes et al., 2020; McJannet et al., 2008;
Tanny et al., 2008). However, a significant challenge arises from the fact that water
possesses a significantly higher heat capacity compared to other land types, resulting in
pronounced heat storage effects in lakes and reservoirs (Jensen, 2010). The thermal
energy within these water bodies tends to move from shallower to deeper regions (Wang



et al., 2023), consequently influencing evaporation, sensible heat flux, and net
longwave radiation losses at the surface. To address biases in evaporation rate
estimations stemming from these factors, Edinger et al. (1968) introduced the concept
of equilibrium temperature. Subsequently, De Bruin (1982) incorporated this concept
into the estimation of evaporation rates. Zhao and Gao (2019) further enhanced
evaporation estimations in open water by establishing a generalized formula for
equilibrium temperature, where water depth emerged as a crucial parameter for
estimating equilibrium temperature.

9        With advancement of remote sensing technology and modern measurement

techniques, lakes and reservoirs have been comprehensive measurement and scrutiny,
spanning from local basins to national and even global scales (Li et al., 2020; Zhang et
al., 2019a). Despite these advancements, existing research primarily utilizes remote
sensing data to calculate changes in surface water area, while there is a notable absence
of accessible data for estimating surface water evaporation losses, such as water depth
information. Consequently, while the Penman equation offers numerous advantages, its
application in hydrological models does have limitations, particularly in accurately
accounting for the complex thermal dynamics of water bodies.

18        The objective of this study is to integrate remote sensing data with an open water

evaporation model to assess evaporation losses on the Loess Plateau. Utilizing a
modified Penman equation, which estimates surface water evaporation based on
equilibrium temperature, and incorporating a water depth calculation, this study
comprehensively accounts for variations in water body characteristics, notably water



depth and surface area. This enhanced methodology aims to achieve a more reliable
estimation of energy fluctuations arising from water body heat storage, thereby
providing a thorough assessment of regional-scale water evaporation losses. This
assessment is crucial for facilitating effective regional or local water resource
management. The primary research objectives are twofold: 1) to estimate the spatial
and temporal variability of surface water evaporation rates and volumes on the Loess
Plateau, and 2) to identify the key driving factors underlying surface water evaporation
losses, with a particular emphasis on the influence of surface water bodies.
**2.  Data and Methods**
**2.1 Study area**

11       The Loess Plateau is located in the northwestern region of China, with an area of

approximately 640,000 km$^2$. Influenced by summer monsoon from the southeast, its
climatic conditions in the area show a gradual change from southeast to northwest.
Annual precipitation shows a decreasing trend following this spatial pattern with an
area-average precipitation of about 440 mm. Meanwhile, seasonal characteristics are
significant, with rain and heat coinciding, rainfall is mainly concentrated in summer in
the form of heavy rainfall (60-70%) (Jiang et al., 2021; Sun et al., 2015). The region is
predominantly located within the Yellow River basin (Fig. 1a), encompassing
subsidiary rivers such as the Wei River. However, the overall availability of surface
water resources is still relatively scarce (Xiao et al., 2019).

21       Characterized by its loose soil structure, the Loess Plateau makes it highly



susceptible to severe soil erosion due to wind and water (Jiang et al., 2019; Zhao et al.,
2013). In order to retain soil and sediment and to reduce the amount of sediment load
to the main channel, a large amount of small-scale check dams have been constructed
within gullies and small tributaries on the Loess Plateau after 2000 (Wang et al., 2021;
Zhang et al., 2022). Driven by the economic development of agriculture, industry, and
various sectors, there has been a heightened demand for water resources. Consequently,
hydraulic infrastructure including reservoirs, has been continuously expanding.
permanent water bodies on the Loess Plateau grow from 1,200 km$^2$ in 2000 to 2,200
km$^2$ in 2020, and the number of small water bodies has increased from 6,721 to 14,082
(Liu et al., 2023). Furthermore, there is a widespread distribution of agricultural
irrigation districts in the western and northern regions of the Loess Plateau, etc.,
Ningxia Irrigation District and Hetao Irrigation District (Zhang et al., 2019). All of these
factors collectively influence the fraction of surface water bodies in the Loess Plateau
(Fig. 1b).

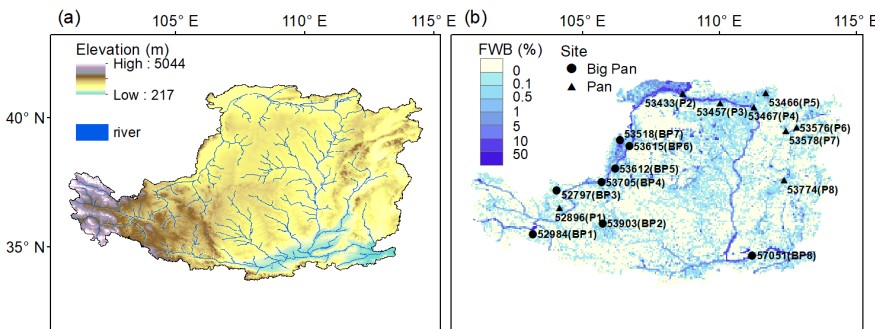

**Fig. 1.** Location of the Loess Plateau and its water body distribution: (a) Elevation and
river systems of the Loess Plateau; and (b) Distribution of evaporation measurement
sites for big pan and pan, and fraction of water body (FWB) on the Loess Plateau.



## 2.2 Data

The data used in this study include remote sensing surface water area, meteorological forcing, and other ancillary information. Monthly surface water area data for the Loess Plateau were obtained from the Joint Research Center (JRC) Global Surface Water dataset (GSW) (Pekel et al., 2016). This dataset utilized an expert system classifier based on Landsat satellite imagery to map the spatial dynamics of global surface water, with a spatial resolution of 30 m. The JRC-GSW data have been successfully applied to detect changes in surface water in the Loess Plateau (Liu et al., 2023). The driving meteorological data used for surface water evaporation estimation include temperature, specific humidity, shortwave radiation, and wind speed, sourced from the China Meteorological Forcing Dataset (CMFD) (He et al., 2020) with a spatial resolution of 0.1°. The CMFD data have undergone comprehensive validation in China with reliable performance (Lei et al., 2023; Tan et al., 2021; Wang et al., 2024; Zhang et al., 2023). Additionally, surface wind speed data from ECMWF Reanalysis 5th Generation (ERA5) (Hersbach et al., 2020) were employed to determine the prevailing wind direction, which was then used to calculate downwind width of water bodies. The ancillary data regarding elevation information were from Advanced Spaceborne Thermal Emission and Reflection Radiometer Global Digital Elevation Model (ASTGTM), available at https://lpdaac.usgs.gov/products/astgtmv003/.This dataset has a horizontal spatial resolution of 30 m and a vertical resolution of 1 m.

We obtained open-water evaporation observations from China Meteorological Administration (http://data.cma.cn/). The observation of the evaporation was





performed using big pan (E601) and small pan with diameter of 20 cm. Given the
difference between the evaporation from the pan and the near water body, a few studies
adjusted the observations using pan coefficients so that the observations is comparable
with the evaporation from the near water body (Jinhui and Zhanbin, 2007; Li et al.,
2018; Sheng et al., 2007). Pan coefficients may vary with the size of the pan and the
seasons. Shi et al. (1986) proposed specific pan coefficients that are suitable for
different regions in China. Based on the study, we set the pan coefficient as 0.95 for the
E601 and 0.75 for the small pan observations, which have been successfully used in
reservoir evaporation estimation in the upper Yellow River (Bai et al., 2023).
**2.3 Evaporation estimation**
**2.3.1 Evaporation rate**
To account for the effect of heat storage in a water body, we proposed a modified
Penman equation that incorporates water depth into the heat storage estimation, thereby
enabling more accurate computation of surface water evaporation (Penman, 1948). This
equation follows the traditional Penman equation, and explicitly considers the heat
storage:

$$E = \frac{s(R_n - \Delta U) + \gamma f(u)(e_s - e_a)}{\lambda_v(s + \gamma)} \tag{1}$$

where $E$ is the evaporation rate ($mm \cdot d^{-1}$), $s$ is the slope of the saturation vapor
pressure curve ($kPa \cdot °C^{-1}$); $R_n$ is the net radiation ($MJ \cdot m^{-2} \cdot d^{-1}$); $\Delta U$ is the heat
storage changes of the water body ($MJ \cdot m^{-2} \cdot d^{-1}$); $f(u)$ is the wind function ($MJ \cdot$
$m^{-2} \cdot d^{-1} \cdot kPa^{-1}$); $e_s$ is the saturated vapor pressure at air temperature (kPa); $e_a$ is
the air vapor pressure (kPa); $\lambda_v$ is the latent heat of vaporization ($MJ \cdot kg^{-1}$); and $\gamma$



is the psychrometric constant ($kPa \cdot {}^\circ C^{-1}$).

2        In the Penman equation, it is assumed that the input meteorological variables are

derived from the surface water. However, due to data limitations, we can only obtain
meteorological data based on land surface observations. To address the errors associated
with the land-based meteorological data, Mcjannet et al., (2012) developed a widely
used wind function:
$$f(u_2) = \lambda_v(2.33 + 1.65u_2)L_f^{-0.1} \tag{2}$$
where $f(u_2)$ is the wind function ($MJ \cdot m^{-2} \cdot d^{-1} \cdot kPa^{-1}$); $u_2$ is the wind speed at
the height of 2 m ($m \cdot s^{-1}$); $L_f$ is the fetch length of the water body (m), fetch length
is the surface water width under prevailing wind direction.

11       Another key factor affecting estimation accuracy arises from variations in the

energy stored within the water body. The introduction of equilibrium temperature serves
as an effective solution (De Bruin, 1982; McMahon et al., 2013). Here, we utilize the
more general equilibrium temperature formula derived by Zhao and Gao (2019):
$$T_e = \frac{[k\varepsilon_a + f(u) \cdot (s+\gamma)] \cdot T_a + (1-\alpha)K{\downarrow} - b(\varepsilon_w - \varepsilon_a) - f(u)(e_s - e_a)}{k\varepsilon_w + f(u) \cdot (s+\gamma)} \tag{3}$$
where $T_e$ is the equilibrium temperature (℃); $\varepsilon_a$ and $\varepsilon_w$ are emissivity of air and
water with cloudiness factor, respectively; $k$ and $b$ are constants of 0.46 $MJ \cdot m^{-2} \cdot$
$d^{-1} \cdot {}^\circ C^{-1}$ and 23.38 $MJ \cdot m^{-2}$, respectively. Based on such equilibrium temperature,
the water temperature can be estimated as
$$T_w = T_e + (T_{w0} - T_e) \cdot e^{\frac{-\Delta t}{\tau}} \tag{4}$$
where $T_w$ is the water column temperature at the current time step (℃); $T_{w0}$ is the
water column temperature at the previous time step (℃); $\Delta t$ is the time step (set as



one month in this study); and $\tau$ is the lag time (d), can be expressed as
$$\tau = \frac{\rho_w c_w \bar{h}}{4\sigma(T_{wb}+273.15)^3 + f(u)(s_{wb}+\gamma)}$$    (5)
where $\rho_w$ is the water density ($kg \cdot m^{-3}$); $c_w$ is the specific heat of water ($MJ \cdot kg^{-1} \cdot$
$°C^{-1}$); $\bar{h}$ is the average water depth (m); $T_{wb}$ is the wet-bulb temperature (°C); and
$S_{wb}$ is the slope of the saturation vapor pressure curve at $T_{wb}$ ($kPa \cdot °C$).

6        The change in the heat storage of water is calculated by the difference in heat

between the moments at the current time and initial time step, using the following
equation:
$$\Delta U = \rho_w c_w \bar{h} \frac{T_w - T_{w0}}{\Delta t}$$    (6)
where $\Delta U$ is the changes of water storage heat.
**2.3.2 Water depth estimation**

12        In the estimation of evaporation, accounting for water depth variation is crucial,

as it fundamentally influences heat storage and transfer dynamics, as shown in
Equations (5) and (6). A significant challenge arises in quantifying evaporation rates
for surface water, largely attributed to the limited availability of comprehensive water
depth data. Digital Elevation Models (DEMs), while valuable, are typically restricted
to capturing surface-level information, thereby hindering the acquisition of detailed
underwater terrain features. To circumvent this limitation, we proposed a water depth
estimation algorithm that operates on the assumption of slope equivalence between the
water body and its boundaries, as depicted in Figure 2. This approach first relies on the
elevation and slope of the land pixels to estimate the water bottom elevation of the
boundary water grids. All land pixels and the calculated water grids are marked as



known. Subsequently, the water bottom elevation of interior grids is iteratively
determined using the same approach based on known neighboring grids within their
eight-neighborhood.

4        For each grid cell (with a resolution of 30 m × 30 m), the determination of water

bottom or bed elevation is expressed as:

$$H_w = \frac{\sum_{i=1}^{n}(H_i - \tan S_i \times D_i)}{n} \tag{7}$$

where, $H_w$ is the water bottom elevation of a target grid cell with resolution of 30 m;
$n$ is the total number of marked as known in the eight-neighborhood; $H_i$ is the $i^{th}$
elevation value of the water body boundary or the already calculated elevation value of
the water grid; $S_i$ is the slope of the $i^{th}$ grid; $D_i$ is the distance from the $i^{th}$ grid to the
target water grid. The calculation for water body grid follows the rule of starting from
the nearest grids to the water body boundary and progressing to the farthest ones.

13       For a given water body, its average water depth is defined as the difference

between the mean elevation of the land boundary grids and the mean waterbed elevation
of the water grids:

$$\bar{h} = \overline{H_b} - \overline{H_w} \tag{8}$$

where $\bar{h}$ is the average water body depth; $\overline{H_b}$ is the average elevation of the water
body boundary; $\overline{H_w}$ is the average elevation of the waterbed.





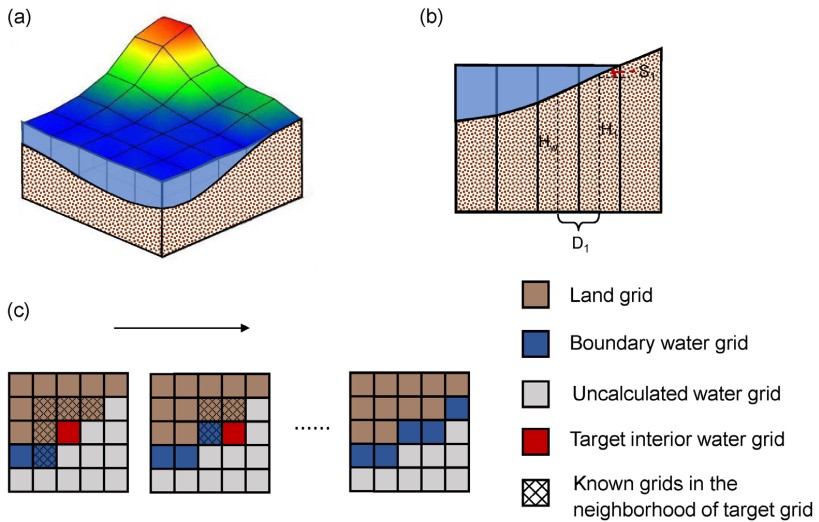

**Fig. 2.** The water bottom elevation calculation: (a) three-dimensional schematic of a
water body, (b) longitudinal section view a water body, and (c) the iterative process of
calculating the bottom elevation of water grids, which involves progressively
computing based on the known elevations of the eight-neighboring grid points and
extending step-by-step into the interior water grids.
**2.3.3 Evaporation volume**
The total evaporation volume (EV) or evaporation loss can be expressed as,

$$EV = E \times Area \times 1000 \tag{9}$$

where $EV$ is monthly average evaporation volume (m$^3$/d), $E$ is monthly average
evaporation rate (mm/d), and $Area$ is surface water bodies area within each grid cell
(km$^2$).
**2.4 Attribution analysis**



The change in evaporation volume of a water body is generally driven by its
surface water area and four climate factors, including air temperature, humidity, wind
speed, and downward shortwave radiation. To quantify the impacts of these factors, we
designed five simulation experiments corresponding to the five factors. In each of the
experiments associated with climate change effect, we detrended one of the four factors
by removing the linear variation of the annual averages, and kept the dynamics of the
other three factors. The detrended equation is presented as

$$M'_{Y_{i,d}} = \left[\frac{M_{Y_i} + \alpha \times (Y_b - Y_i)}{M_{Y_i}}\right] \times M_{Y_{i,d}} \qquad (10)$$

where $Y_i$ is one year from 2000 to 2018; $M'_{Y_{i,d}}$ is the daily-scale meteorological
forcing variable (including temperature, wind speed, specific humidity, and shortwave
radiation) for $Y_i$ after detrend; $M_{Y_{i,d}}$ is the original daily-scale meteorological data;
$M_{Y_i}$ is the annual average meteorological data for the year $Y_i$; $Y_b$ is the reference year
in 2000, and $\alpha$ is the trend in the annual average meteorological data. For the
experiment associated with the surface water area, the same equation can be used for
detrending the surface water area. However, since the surface water area is based on
monthly-scale data, $M'_{Y_{i,d}}$, and $M_{Y_{i,d}}$ respectively denote the detrended monthly-scale
water area and the original water area, and the parameter of $\alpha$ represents the trend
derived from the monthly-scale original water area. This equation is able to remove the
annual trend but preserve the seasonal variation. Based on the five experimental
simulations and the base simulation that estimate the long-term evaporation volume in
the study period, we can calculate the contribution of each factor as,

$$Con_x = \frac{Trend_{E_x} - Trend_{E_{base}}}{Trend_{E_{original}}} \times 100\% \qquad (11)$$



where $Con_x$ is contribution percentage of the variation in element $x$ (e.g.,
temperature, specific humidity, shortwave radiation, wind speed, and surface water area)
to the changes in $EV$; $Trend_{E_x}$ is the trend of $EV$ after detrending all factors except
for the element $x$; $Trend_{E_{base}}$ is the trend of $EV$ after detrending all factors; and
$Trend_{E_{original}}$ is the original trend of $EV$.

## 3. Results

### 3.1 Evaluation of evaporation rate estimation

We first evaluate the evaporation estimation using the observations that have been
adjusted as described in subsection 2.2. Figure 3 illustrates a comparison between the
estimated evaporation from water bodies and the observations from the small-pan
evaporation. The evaporation estimates exhibit a strong agreement with the
observations, capturing the monthly dynamic changes. The overall coefficient of
determination ($R^2$) for this comparison is 0.75, indicating a robust correlation. The bias
is minimal, with a value less than 5 mm/mon, and the root mean square error (RMSE)
stands at approximately 22.54 mm/mon. However, at certain stations, such as station
P6, the modified Penman equation slightly underestimates peak evaporation values.
Nevertheless, the overall alignment is deemed acceptable.





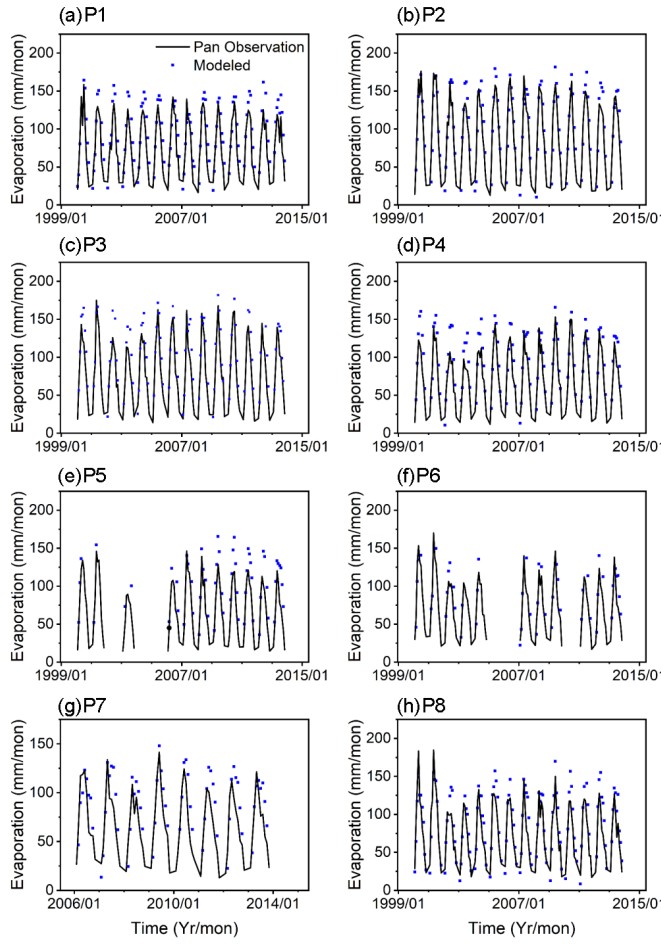

**Fig. 3.** Comparison of the estimated evaporation and the observations from small pan

at eight stations: the plots from (a) to (h) are for the eight stations as shown in Figure 1.

4  Figure 4 provides a comparison between the estimated evaporation and the

observations recorded by big pans. This comparison underscores a notable alignment,
reflecting a coefficient of determination ($R^2$) of 0.66. This strong correlation indicates
that the model effectively captures the temporal variations in water body evaporation.
Importantly, the performance metrics for the big-pan evaporation show an improvement



over those from the small pans. Specifically, the bias in the estimation for the big pans
is lower, remaining well within 1 mm/mon, and the root mean square error (RMSE)
decreases to 22.50 mm/mon, compared to the RMSE for the small pans in Figure 3.
This enhanced accuracy can be attributed to several factors. Big evaporation pans, by
their design, cover a greater surface area and thus provide a more representative
measure of evaporation from larger water bodies. Their size likely mitigates the
influence of localized environmental variabilities, such as temperature fluctuations and
wind patterns, which can disproportionately affect smaller pans. Furthermore, the closer
resemblance of big evaporation pans to actual water bodies in terms of surface area and
heat exchange dynamics may contribute to their higher observational accuracy. This
similarity likely reduces systematic errors and improves the overall agreement between
simulated and observed evaporation rates.



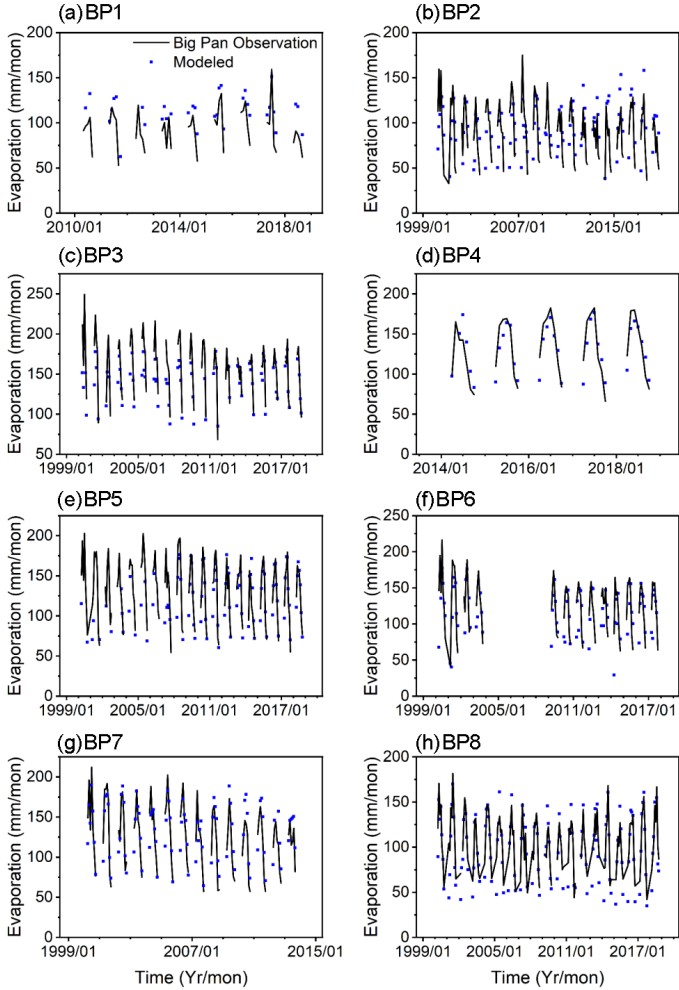

**Fig. 4.** Comparison of the estimated evaporation and the observations from big pan at
eight stations: the plots from (a) to (h) are for the eight stations as shown in Figure 1.

4       Figure 5 presents the spatial distribution of $R^2$, Bias, and RMSE, offering a

nuanced understanding of the model's performance across different regions. Notably,
the upstream areas exhibit larger estimation errors, with bias exceeding 5 mm/mon at
three stations. This could be attributed to the complex terrain and lower temperatures





in these regions, which might introduce significant uncertainties in water depth
calculations, subsequently affecting the accuracy of evaporation estimates. In contrast,
the midstream and downstream regions demonstrate better accuracy, likely due to more
homogeneous environmental conditions and milder temperature variations. The overall
consistency between the simulated evaporation and the observations underscores the
reliability of the modified Penman method for estimating evaporation, despite some
localized discrepancies. This confirms the method's applicability for analyzing
spatiotemporal variations in water body evaporation across the diverse landscapes of
the Loess Plateau region.

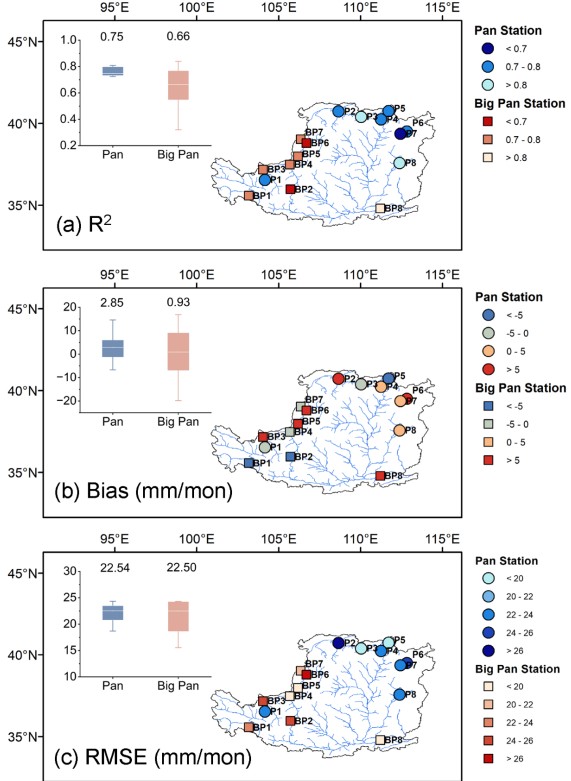





**Fig. 5.** The spatial distribution and box plots of the three metrics (a) coefficient of

determination ($R^2$), (b) bias, and (c) root-mean-square error (RMSE) for simulated data

comparing to pan and big pan.

## 3.2 Spatial-temporal variation in evaporation rate

Evaporation from water bodies across the Loess Plateau demonstrates pronounced

geographical heterogeneity. The spatial distribution of evaporation exhibits a gradual

decrease from the northwest to the southeast, as depicted in Figure 6a. The long-term

average evaporation rates vary between 2.8 and 3.1 mm/d, with certain areas in the

northwest exceeding 5 mm/d. This spatial pattern suggests that regional climate

conditions, including temperature, humidity, and wind speed, significantly influence

evaporation rates. Temporal changes in evaporation, however, do not follow a

discernible trend and appear relatively decentralized, as illustrated in Figure 6b. Despite

this, areas experiencing decreasing evaporation are slightly more extensive than those

with increasing evaporation. Notably, regions with significant increases in evaporation

rates ($p < 0.01$) are concentrated in the western and north-central parts of the Loess

Plateau, indicating localized factors may be driving these increases, such as changes in

water body characteristics.

Figure 6c and d present the interannual change and seasonality of evaporation rates.

Over the period from 2000 to 2018, the long-term average daily evaporation rate on the

Loess Plateau was approximately 2.98 mm/d. However, this rate decreased slightly at

a rate of −0.0031 mm/d/yr, indicating a subtle but consistent decline in evaporation over



the study period. The interannual variability in evaporation is substantial, with notable
lows in 2003, when daily evaporation rates dropped to approximately 2.8 mm/d. This
year likely experienced unusual climatic conditions that suppressed evaporation, such
as increased cloud cover, reduced temperatures, or higher than average precipitation. In
contrast, other years maintained average evaporation rates around 2.98 mm/d, reflecting
the typical evaporative conditions of the region. Please note December and January
were not considered for evaporation estimation in this study due to low temperatures
and freezing of the water bodies.

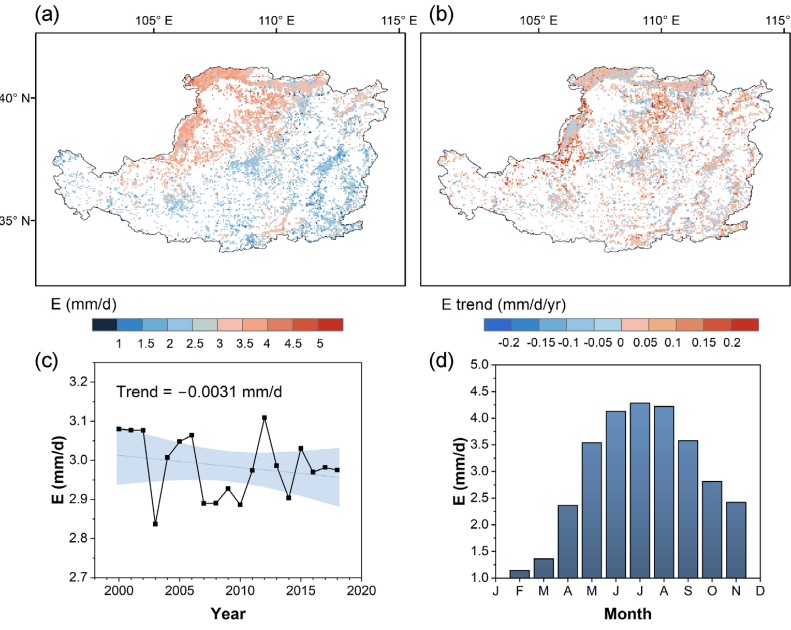

**Fig. 6.** The spatial distribution of (a) long-term average daily evaporation rate and (b)
temporal trends. Average variations in evaporation rates for (c) annually dynamics and
(d) monthly climatology.



### 3.3 Spatial-temporal variation in evaporation volume

The spatial distribution of evaporation volume across the Loess Plateau is depicted in Fig. 7a, revealing a distinct pattern of high evaporation in the northwest and lower values in other regions. In the densely watered northwestern areas, evaporation volumes exceed 20,000 $m^3$/d, contrasting sharply with most regions where evaporation remains below 1,000 $m^3$/d. A notable increasing trend in evaporation volume is observed, particularly concentrated in the northwest (Fig. 7b), coinciding with areas experiencing high evaporation loss. Additionally, significant changes in surface water evaporation loss are evident in the southeastern and central-northern parts of the plateau, where the evaporation volume increased with the rate up to 100 $m^3$/d/yr.

Figure 7c illustrates the temporal change in daily evaporation volume on the Loess Plateau. A significant upward trend ($P < 0.01$) is observed, with an average annual increase of $0.117\times10^6$ $m^3$/d/yr. Evaporation volumes rose from $3.18\times10^6$ $m^3$/d in 2000 to $5.69\times10^6$ $m^3$/d in 2018, with a long-term average of $4.16\times10^6$ $m^3$/d for the period 2000-2018. Seasonal variation in evaporation volume exhibits distinct peaks in spring and autumn, with a yearly maximum of $6.04\times10^6$ $m^3$/d in May. It is noteworthy that while evaporation rates peak during the summer, evaporation volumes peak in May, aligning with seasonal fluctuations in water body areas (Liu et al., 2023). This implies that the seasonal variation in evaporation volume may be dominated by changes in surface water area.

An analysis of temporal fluctuations reveals inconsistencies between evaporation



loss and evaporation rate. Specifically, years with low evaporation loss, such as 2000,
2001, 2003, and 2011, do not always correspond to years with low evaporation rates,
which were 2003 and 2010. This discrepancy also suggests that factors other than
meteorological conditions may control evaporation loss. The difference in patterns
suggests that while evaporation rates are largely driven by climatic factors like
temperature, humidity, and wind speed, evaporation loss may be more sensitive to
changes in water body characteristics, such as surface area, depth, and water availability.

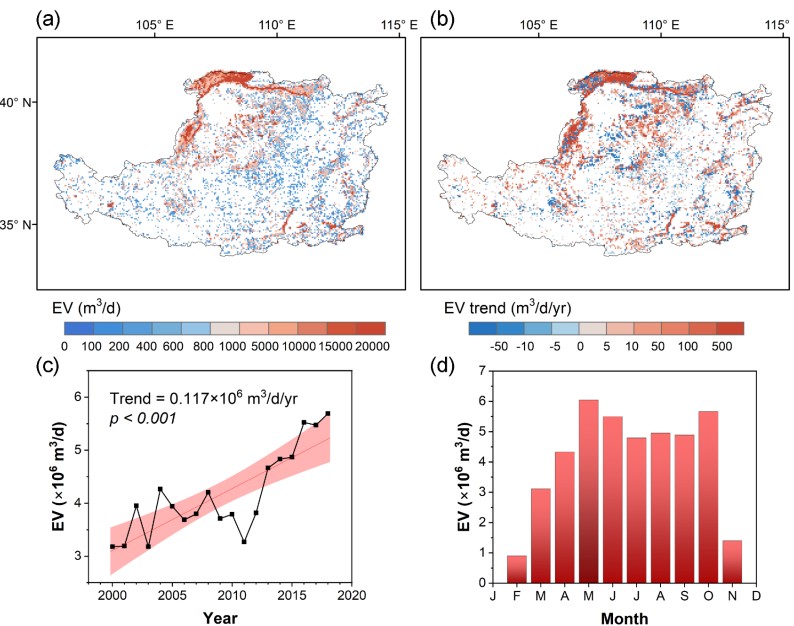

**Fig. 7.** The spatial distribution of (a) average evaporation volumes and (b) their
temporal trends. Average variations in evaporation volumes for (c) annually and (d)
monthly climatology.
**3.4 Effect of driving factors on evaporation loss**



### 3.4.1 Dynamics of driving factors

Under the influence of global warming and intensified human activities, the meteorological factors across the Loess Plateau have undergone significant transformations over the past two decades. An analysis of their spatial and temporal variations reveals complex patterns. Air temperature variations across the region depict a predominantly increasing trend, with notable exceptions along the northwest-southeast median and a few isolated areas where decreases are observed (Fig. 8a). The southern-central part of the plateau experiences the most substantial cooling, exceeding −0.06 °C /yr, while the western region undergoes the fastest warming, reaching rates of up to 0.1 °C/yr. Over the past two decades, the mean annual temperature on the Loess Plateau has stabilized around 9.32°C, albeit with inter-annual fluctuations indicating a slight increase of 0.02 °C/yr ($p < 0.1$) (Fig. 8b).

Specific humidity follows a similar pattern of increase across most of the plateau, contrasted with a decreasing trend in a small southeastern sector (Fig. 8c). Over the study period, the annual average specific humidity initially declined slightly during the first decade, subsequently experiencing a rapid increase, with an average rate of $1.29 \times 10^{-5}$ kg/kg/yr ($p < 0.1$) (Fig. 8d). In contrast, surface shortwave radiation exhibits a marked decreasing trend across the majority of the region, with only a small eastern-central area showing an upward trajectory (Fig. 8e). The annual average surface shortwave radiation demonstrates a fluctuating but overall downward trend, decreasing at a rate of −0.18 W/m$^2$/yr (Fig. 8f). Although the number of water grids with increasing wind speed is comparable to those with decreasing wind speed, the magnitude of the





increase is approximately three times larger than the decrease. Hence, the annual
average wind speed displayed a clear upward trend, with a mean increase rate of 0.02
m/s/yr ($p < 0.05$) (Fig. 8h).

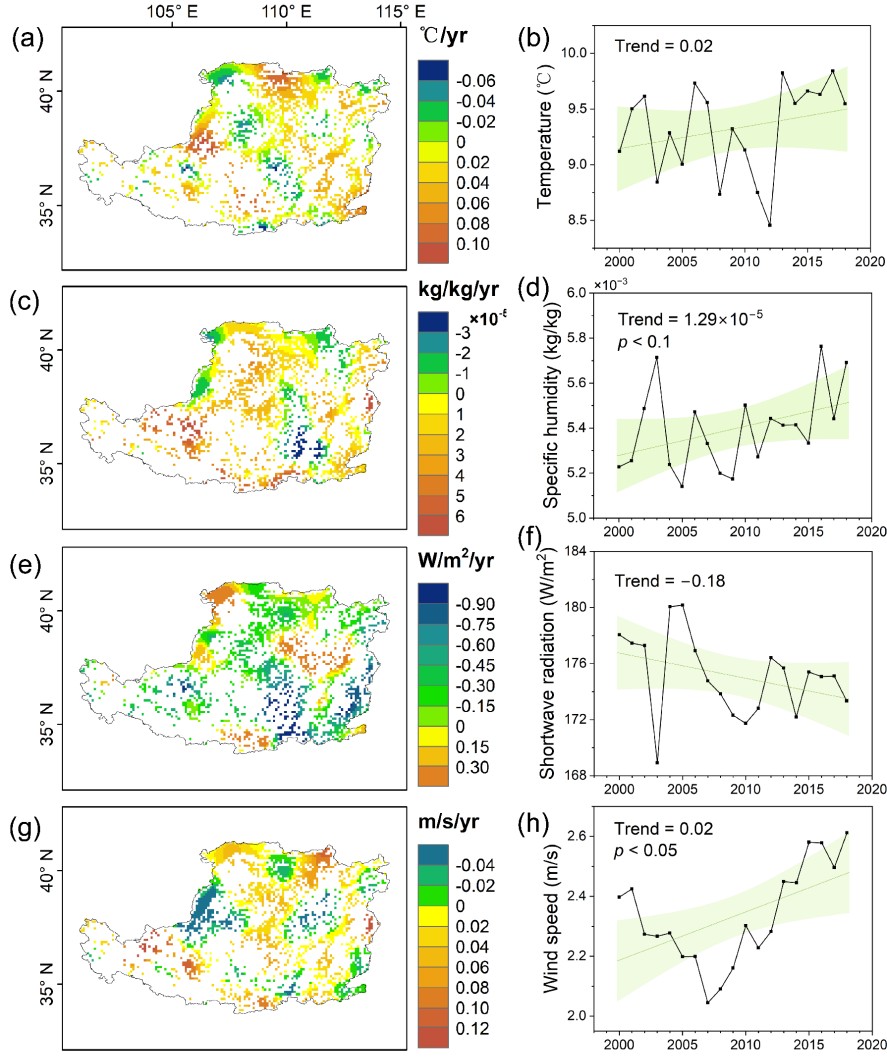

**Fig. 8.** Average annual trends distribution of (a) temperature, (c) specific humidity, (e)
surface shortwave radiation, and (g) wind speed. Trends in average annual time in (b)



temperature, (d) specific humidity, (f) surface shortwave radiation, and (h) wind speed.
Compared with meteorological factors, surface water area typically exhibits more
pronounced variations in areas of high human activity. As shown in Figure 9, the surface
water bodies of the Loess Plateau are concentrated in the northwestern region, while
the rest of the region exhibits a more dispersed distribution. Notably, water bodies
smaller than 0.05 km$^2$ account for 73.8% of all water body grids on the Loess Plateau
(Fig. 9a). Among these, grids showing an increasing trend are nearly 1.5 times those
with a decreasing trend (Fig. 9b). This substantial rate of increase underscores the
dynamic nature of water body expansion in the region. In parallel, there is a more
pronounced trend of growth in the northwestern and central regions, further
emphasizing the significant changes in surface water area over time. This pattern is
consistent with the distribution characteristics of evaporation loss as depicted in Figure
7. Such changes have important implications for the evaporation loss and water
resource allocation, warranting continued evaluation and research.



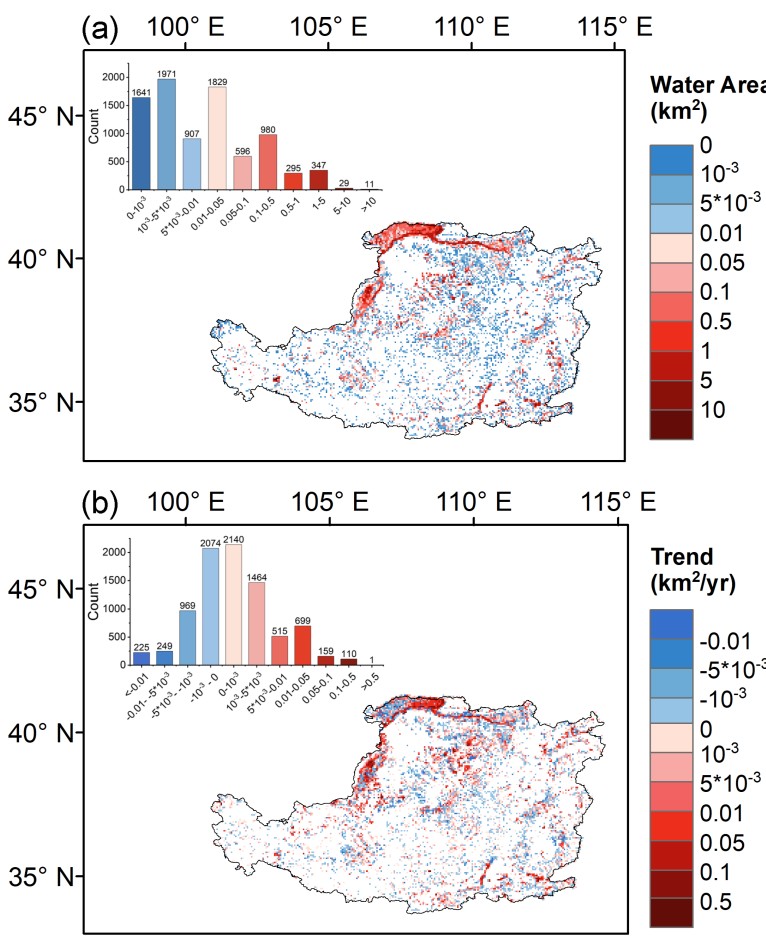

**Fig. 9.** Average distribution and grid count from 2000 to 2018 of (a) water area for 0.05°
gird and (b) the trend of water area.

### 3.4.2 Contribution of driving factors

To quantify influence of each factor on the evaporation loss, we designed five
detrend experiments that are associated with the water area and the four meteorological
factors as described in subsection 2.4. As shown in Figure 10, the result reveals that
water area variations play a significant role in modulating evaporation processes.



Across the study area, changes in water surface area account for a substantial 101.12%
of the variation in evaporation volumes, indicating a strong and positive effect on
evaporation dynamics (Fig. 10a). Spatially, this effect is particularly pronounced on the
Loess Plateau, where water area changes contribute to more than 50% of the
evaporation volume variations in most water bodies (see Fig. 10b). These changes
largely enhance evaporation loss, underscoring the critical role of water body dynamics
in regulating evaporation at both local and regional scales. Notably, the central part of
the Loess Plateau exhibits a noticeable decreasing trend in evaporation volumes, which
can be attributed to the shrinkage of water bodies in this area.
In contrast, the influence of meteorological factors on evaporation volumes is
relatively modest. Among these factors, shortwave radiation exhibits the most
significant effect, with a contribution of 0.33%. However, the cumulative effect of all
meteorological factors only accounts for 0.5% of the variation in evaporation volumes,
suggesting potential offsetting trends among these factors. Spatially, the contributions
of meteorological factors to evaporation volumes are either positive or negative but
remain relatively small, with specific humidity contributing below 5% in most regions
and the other three meteorological factors generally contributing less than 10% (Figure
10c-f). The above results emphasize the primary importance of water area dynamics in
regulating evaporation volumes, while meteorological factors play a secondary, albeit
complex role.

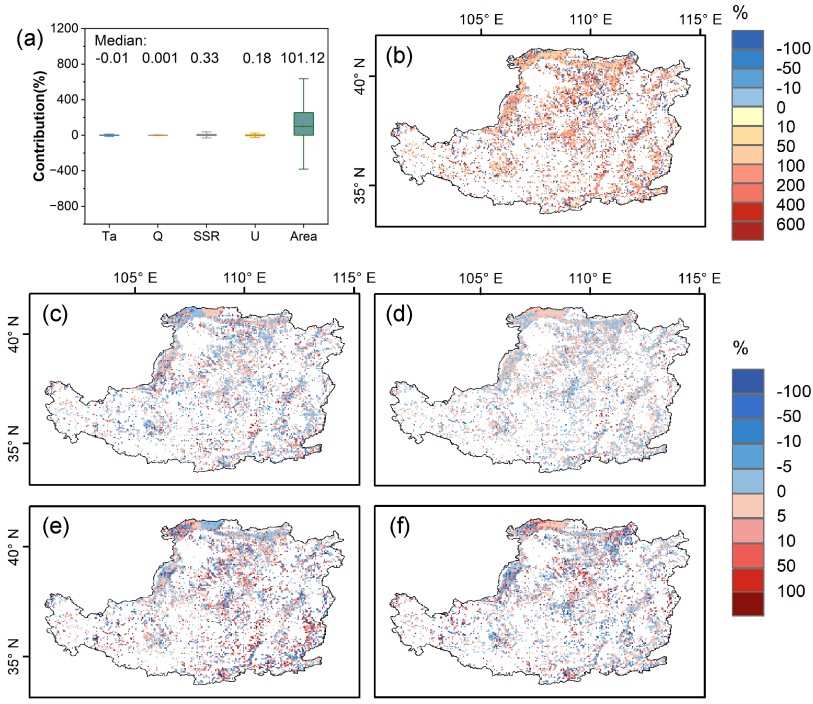

**Fig. 10.** (a) The contributions of factors to the changes in evaporation volume. The factors considered are air temperature (Ta), specific humidity (Q), surface shortwave radiation (SSR), wind speed (U), and water area (Area). Changes in evaporation volume caused by (b) water area, (c) air temperature, (d) specific humidity, (e) surface shortwave radiation, and (f) wind speed.

## 4. Discussion

### 4.1 Reliability of evaporation estimation

A significant challenge in estimating evaporation rates arises from fluctuations in





the heat storage of the water body. The thermal storage of a water body directly
influences its energy contribution and water surface temperature (McMahon et al.,
2013), subsequently impacting evaporation rates (Fairall et al., 1996; Leconte, 2015;
Nehorai et al., 2013). For instance, the incorporation of the thermal storage term in Lake
Mead (Nevada/Arizona) improved the estimated $R^2$ from 0.29 to 0.84 compared to
models that excluded it (Zhao and Gao, 2019). Furthermore, water depth emerges as a
critical parameter in estimating the heat storage capacity of a water body. In particular,
the median depth in China (31.8 m) is significantly greater than that in the United States
(21.9 m) (Tian et al., 2021). Deeper water bodies, particularly when there is a substantial
temperature difference between air and water, possess a greater heat storage capacity.
This capacity further moderates the disparity between air temperature and water surface
temperature, a factor crucial for accurately estimating the heat storage dynamics related
to water body depth.

14        Incorporating these dynamics is important to provide a more refined estimation of

surface water evaporation (Panin et al., 2006; Wossenu, 2001; Zhang et al., 2024). To
consider the effect of heat storage on evaporation, we developed a modified Penman
model that incorporates the concept of equilibrium temperature, which was successfully
used to estimate the evaporation rate of surface water bodies across Loess Plateau. The
comparison between the simulated evaporation rates and those measured using
evaporation pans reveals a coefficient of determination ($R^2$) of approximately 0.7, with
a relative bias of less than 5 mm/mon. This level of agreement underscores the
robustness of our model and its capacity to accurately replicate observed evaporation

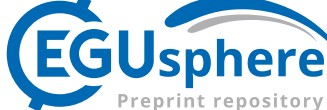

patterns. Importantly, the averaged evaporation rate estimated for the Loess Plateau in
this study is about 2.98 mm/d, demonstrating excellent consistency with evaporation
rates reported for specific reservoirs or water bodies in the region. For instance,
previous studies focusing on various water bodies within the Loess Plateau have
generally reported evaporation rates falling within the range of 2.73 to 3.72 mm/d
(equivalent to 1000~1358 mm/yr) (Ma Haijiao et al., 2013; Ren and Guo, 2006; Tlan
et al., 2005). This alignment indicates that our methodology, which integrates multiple
physical parameters, is capable of producing evaporation estimates that are in good
agreement with those derived from more localized studies.
Particularly noteworthy is the high evaporation rate observed in the northwest
region of the Loess Plateau, reaching 4~5 mm/d. This finding is corroborated by similar
observations from Ding et al. (2012), who reported an evaporation rate of
approximately 4.17 mm/d in the same area. The consistency across these studies,
despite differences in methodologies and study areas, suggests that our approach
captures the spatial variability of evaporation rates within the Loess Plateau, likely
reflecting regional differences in climate and surface water coverage.
To further solidify the estimated evaporation loss, we delved into an analysis of
evaporation volume, while few studies have estimated this variable for water bodies in
the Loess Plateau. Among the limited research available, Zhang et al., (2014) estimated
the evaporation loss from the reservoir upstream of Huayuankou at approximately 4.14
$\times 10^6$ m$^3$/d (equivalent to $1.51 \times 10^9$ m$^3$/yr). In comparison, our study calculated an
evaporation loss of $4.16 \times 10^6$ m$^3$/d, a value closely aligned with Zhang et al.'s findings.





This similarity provides a validation of our estimation methodology, suggesting that
our calculated evaporation loss for the Loess Plateau reservoirs is both reasonable and
reliable. By demonstrating consistency with established results, our findings offer
further confidence in the applicability of our methods to similar hydrological contexts
in the region.

## 4.2 Evaporation variability and its drivers

This study presented the spatiotemporal variations in water body evaporation
across the Loess Plateau. Our findings unveil a distinct spatial pattern, with evaporation
rates consistently higher in the southeastern region compared to the northwestern part.
This spatial distribution aligns closely with the regional climate gradient (Figures 6 and
8), suggesting a strong influence of climatic conditions on evaporation dynamics. The
relatively higher evaporation rates in the northwest can be attributed to stronger surface
shortwave radiation, lower humidity, and potentially higher wind speeds, all of which
favor enhanced evaporation. When examining temporal trends over the past two
decades, our results indicate a subtle yet non-significant decreasing trend in water body
evaporation rates across the Loess Plateau. This trend is primarily driven by changes in
key climatic factors. Specifically, while air temperature and wind speed have shown an
increasing trend (Figure 8), which typically enhances evaporation, these effects have
been counterbalanced by concurrent increases in air humidity and decreases in solar
radiation. The combined impact of these changes has led to a net, albeit modest,
decrease in evaporation rates.



As to the evaporation volume, our study unveils a notable increase over the Loess
Plateau during the past two decades, with an ascending rate of $0.117 \times 10^6$ m$^3$/d/yr.
Furthermore, there is a distinct seasonal pattern in the evaporation volume, with peaks
occurring in May and October. The observed upward trend in total water evaporation
can be primarily attributed to the expansion of water bodies within the study area. This
expansion is a direct consequence of the escalating human demand for water across
various sectors, including agriculture, industry, domestic use, and ecological
preservation (Liu et al., 2023). To meet these burgeoning needs, numerous reservoirs
and dams have been constructed, leading to an enlargement of surface water.
Additionally, the proliferation of small check dams, aimed at reducing sediment load in
river channels, has further contributed to the augmentation of water body areas. These
anthropogenic interventions collectively facilitate increased water evaporation losses.
Previous studies have explored water evaporation dynamics across various global
regions, revealing significant variations influenced by both natural and anthropogenic
factors. A notable observation is the substantial increase in lake evaporation rates
worldwide, attributed to a 58% rise in evaporation rates coupled with a 23% reduction
in lake ice cover (Zhao et al., 2022). This underscores the profound impact of climate-
related changes, such as rising temperatures and altered precipitation patterns, on lake
evaporation. Furthermore, the proliferation of large reservoirs, particularly in middle-
income countries, has emerged as the primary driver of increased reservoir evaporation
globally (Tian et al., 2022). This highlights the significant role of anthropogenic
interventions, particularly water infrastructure development, in shaping evaporation



trends. Turning to specific regions, studies in the Namoi catchment have shown a
decreasing trend in total evaporation volumes, despite an increasing trend in surface
water evaporation rates. This apparent contradiction has been linked to a reduction in
the frequency of surface water occurrences, indicating that the availability and
persistence of water bodies play a crucial role in modulating evaporation rates (Fuentes
et al., 2020). Similarly, in China, increased evaporation losses from reservoirs have
been attributed to both higher evaporation rates and the expansion of reservoir areas
(Tian et al., 2021).

9        While these studies have contributed to our understanding of water evaporation,

they have primarily focused on large reservoirs or lakes. In contrast, our study
encompasses a comprehensive analysis of all water bodies, including various small-
scale reservoirs and check dams, as well as large reservoirs/lakes in the Loess Plateau
region. To detect the evaporation in large reservoirs in the Loess Plateau, we estimated
48 large reservoirs documented in the GRand database (Lehner et al., 2011,
http://globaldamwatch.org/). As shown in Figure 11, the evaporation rate has a slight
decline for the 48 large reservoirs between 2000 and 2018, accompanied by a decrease
in the total evaporation volume ($-0.29$ m$^3$/d/yr). This trend of the evaporate rate for the
large reservoirs aligns with the average evaporation rate over the Loess Plateau (Figure
6), but contrasts with the increasing total evaporation volume observed across the entire
region (Figure 7). This discrepancy suggests that small- and medium-sized water bodies
significantly contribute to the overall evaporation on the Loess Plateau. The contrasting
trends between large reservoirs and the broader Loess Plateau highlight the complexity



of evaporation dynamics in different water body types and scales.

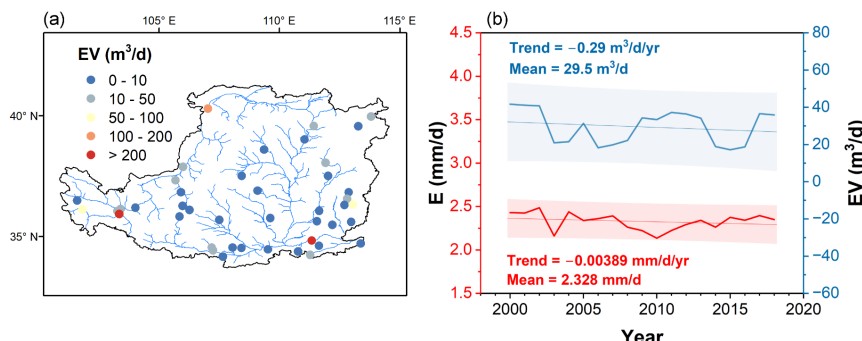

**Fig. 11.** (a) The spatial distribution of 48 large reservoirs and their average evaporation
volume. (b) Temporal evaporation rate (E) and evaporation volume (EV) of the 48 large
reservoirs for the period of 2000 to 2018. The shaded area represents 95% prediction
bands.
**4.3 Implications and limitations**

8        Our study, which employs a modified Penman equation that incorporates dynamic

water depth and surface areas, represents a significant advancement in accurately
estimating evaporation rates and volumes for open water bodies. This methodological
refinement underscores the importance of considering both meteorological factors and
the dynamic nature of water bodies, particularly for small- to medium-sized entities.
Ignoring these dynamics, especially for small- and medium-sized water bodies, can lead
to substantial uncertainties in evaporation assessments, with potential ramifications for
regional water balance calculations (Dawidek et al., 2014; Stan et al., 2016).

16       A notable finding from our research is the paradoxical trend of decreasing



evaporation rates yet increasing total evaporation volumes on the Loess Plateau. This
finding has profound implications for water resource planning and management in the
region. The extensive construction of various reservoirs globally over the past few
decades aimed at augmenting local water supplies for agricultural irrigation, industrial,
or domestic purposes (Baldassarre et al., 2018; Resources, 2006). However, the
development of the water conservancy projects has potential of exacerbating
evaporation losses—a factor that has largely been overlooked in resource planning. By
comparing water evaporation volumes with surface water withdrawal by residents in
the Loess Plateau (Figure 12), we reveal a striking similarity: the magnitude of total
evaporation loss is comparable with the average annual surface water withdrawal
(approximately $1.55 \times 10^9$ m$^3$/yr). The ratio of evaporation to the withdrawal has
escalated from 80% in 2000 to 130% in 2018, highlighting considerable evaporation
loss and a significant threat to water security in the region. Therefore, future water
project planning needs to incorporate evaporation losses to mitigate potential water
resource risks.

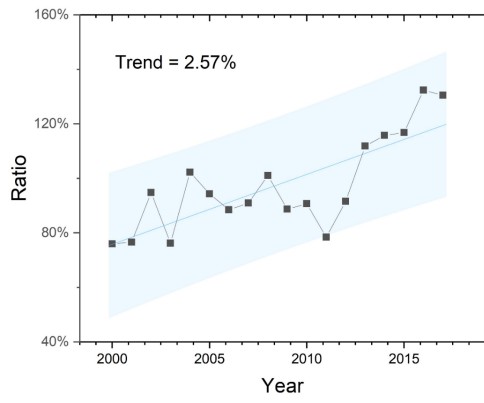



**Fig. 12.** The ratio of surface water evaporation volume to annual average water
withdrawal by residents on the Loess Plateau from 2000 to 2017.
Previous studies generally focus on large reservoirs or lakes to monitor and
investigate their water budgets (Tian et al., 2021, 2022). This may have inadvertently
underestimated the contribution of smaller water bodies to overall evaporation as
evidenced in this study. These smaller entities are often more sensitive to local climatic
and anthropogenic impacts, necessitating a more granular analysis in future research.
To comprehensively understand and manage water resources, it is imperative to extend
monitoring and modeling efforts to small- and medium-sized water bodies, which our
findings suggest play a pivotal role in the regional evaporation budget.
Despite the insights for the methods and findings in our study, several limitations
merit acknowledgment. One notable limitation of our study pertains to the reliance on
the JRC-GSW water body data for calculating water depth and surface area. While this
dataset has been instrumental in our analysis, we did not conduct an independent
assessment of its accuracy and completeness. It is important to highlight that the JRC-
GSW data exhibit seasonal gaps (Liu et al., 2023; Pekel et al., 2016), which could
introduce uncertainties into our calculations of water depth, as well as subsequent
estimations of evaporation rates and volumes. These data absences might reflect
variations in water levels and extents that are not captured by our methodology, thereby
affecting the precision and reliability of our findings. Future research could benefit from
incorporating additional data sources or employing advanced remote sensing
techniques to validate and complement the JRC-GSW dataset, ensuring a more robust



representation of water body dynamics across different seasons.
Another limitation concerns the evaluation of water body evaporation, which was
assessed using evaporation pan data collected near the reservoir. Although this approach
provided a practical strategy for validation, it is acknowledged that evaporation rates
from pans can significantly diverge from those of various water bodies due to
differences in surface characteristics, heat capacity, and exposure to environmental
factors. Although adjustments were made to align pan measurements with actual water
evaporation conditions, a certain level of uncertainty persists in this extrapolation. To
mitigate this limitation and enhance the accuracy of evaporation estimates, future
studies should prioritize the deployment of more comprehensive observational
networks. This could include installing eddy covariance systems at multiple reservoir
levels or employing floating evaporation pans directly on water surfaces to capture
more representative evaporation rates. Such methodologies would not only reduce the
inherent uncertainties associated with current measurement techniques but also provide
a finer spatial and temporal resolution of evaporation processes, ultimately leading to
more accurate and reliable model outputs.

## 17  5.   Conclusions

This study improved the Penman equation to estimate open water evaporation in
the Loess Plateau by incorporating an equilibrium temperature approach, with
consideration of the variations in surface water area and depth. The improved
methodology, validated against adjusted pan evaporation measurements, demonstrated





a robust performance with a coefficient of determination exceeding 0.7 and nearly all
biases below 15 mm/mon, highlighting its efficacy in simulating evaporation dynamics.
Our findings reveal that the average evaporation rate in the Loess Plateau is about
2.98 mm/d over the past two decades, with peak values occurring in May and October,
albeit showing a slight decreasing trend. However, the total evaporation volume or loss
stands at $4.16 \times 10^6$ m$^3$/d, exhibiting a rapid increase at a rate of $0.117 \times 10^6$ m$^3$/d/yr.
Attribution analysis further elucidates that the primary driver behind the changes in the
total evaporation volume is the expansion of water surface area, accounting for a
dominant contribution of the variation, while climatic factors play a minor role.
Particularly, the proliferation of small- to medium-sized reservoirs and check dams in
the Loess Plateau has significantly amplified evaporation losses, which are roughly
equivalent to the annual surface water withdrawal in the region.
These findings underscore the importance of considering the dynamic aspects of
water surface area and depth in assessing the thermal storage capacity of water bodies
for accurate evaporation estimation. Moreover, they emphasize the necessity of
accounting for evaporation losses in water resource management, particularly in the
context of reservoir construction and operation. Enhanced monitoring and estimation
of evaporation losses from small- to medium-sized reservoirs are crucial to bolster
water security in arid regions like the Loess Plateau. This study thus demonstrates that
the research methods employed are readily extendable to other regions. More
importantly It contributes novel insights into the intricate relationship between water
body dynamics and evaporation, with implications for sustainable water resources



planning and management in the face of climate variability and development of the
hydraulic projects.
**Data availability**
The datasets utilized in this study are publicly available from their respective
official sources: The Joint Research Center (JRC) Global Surface Water dataset (GSW)
can be accessed through Google Earth Engine (GEE) platform
(https://doi.org/10.1038/nature20584). The China Meteorological Forcing Dataset
(CMFD) was obtained from the National Tibetan Plateau Data Center
(https://doi.org/10.11888/AtmosphericPhysics.tpe.249369.file). ERA5 monthly data
were downloaded from the Copernicus Climate Data Store
(https://doi.org/10.24381/cds.f17050d7). The Advanced Spaceborne Thermal Emission
and Reflection Radiometer Global Digital Elevation Model (ASTER GDEM) version
3 data were acquired from NASA's Land Processes Distributed Active Archive Center
(https://doi.org/10.5067/ASTER/ASTGTM.003). Meteorological station evaporation
records were provided by the China Meteorological Administration
(http://data.cma.cn/). The data generated in this study (evaporation rate and evaporation
volume) can be accessed in a Zenodo repository (https://zenodo.org/records/14963640).
**Author contributions**
X.X. and Y.L. designed the study. Y.L. carried out image data processing. Y.L. and
X.X. led interpretation of the results and writing of the manuscript. Y.W., A.T., D.P.,



and X.W. contributed to the discussion.

## 2  Competing interests

The authors declare that they have no conflict of interest.

## 4  Financial support

This study is supported by a grant from the National Natural Science Foundation
of China (No. 42271021).

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
