# Peer review of "Increased surface water evaporation loss induced by"

_EGUsphere, 2025_

## Author Comment (AC1)

**Response to Reviewers**

**Note:**

**(1) In this response, the text in *italic type* is the original comments from the reviewers, and the text in** blue, headed with "Reply**", is the response from the authors.**

**(2) In the manuscript, the words in** blue **indicate the sentence is improved or revised. Some of them are mentioned in this response via the page and line number.**

***Summary:***

*The objectives of the study are well described, and the results are thoroughly discussed. A key finding of this study is the paradoxical behavior of decreasing evaporation rates, yet with increasing evaporation volume on the Loess Plateau, whereby mainly the increase in area in small- to medium-sized water bodies over the last decades contributed to this increase in evaporation volume.*

*Below, I provide several comments and suggestions that may help improve the impact of the manuscript.*

**Reply Summary:**

We sincerely appreciate the reviewer's time and effort in evaluating our manuscript, as well as their positive feedback regarding the clarity of the study's objectives and the thorough discussion of the results. The constructive comments will undoubtedly help improve the impact and quality of our work. In response to the reviewer's observations, we will focus on the following revisions in the updated manuscript:

- ➤ Methodological Validation: We will further validate the improved evaporation calculation method, particularly addressing the influence of water depth on evaporation rates, as suggested.
- ➤ Clarity and Precision: We will carefully revise any ambiguous or unclear

descriptions to ensure the findings are presented without misinterpretation.

Below, we provide point-by-point responses to the reviewer's specific comments.

***comments and suggestions:***

*1) • Be cautious when drawing conclusions regarding increases in total evaporation losses over the region, due to the expansion of water body area. Take into account that with an increased extent of water body area over time, the open water evaporation amount will indeed rise. This relationship should be clearly acknowledged when interpreting the results.*

**Reply:** We appreciate the reviewer's insightful comment regarding the potential influence of expanded water body area on total evaporation losses. We will clarify in the Discussion/Conclusion section that the rise in total evaporation loss includes contributions from both climatic factors (e.g., temperature, wind speed) and the expansion of water bodies. We believe these revisions will provide a more balanced interpretation of the findings.

*2) • In the introduction it is highlighted that terrestrial evaporation increased over the past years, where some key climatic factors play a crucial role. One would expect that these variables also induce rises in open water evaporation rates over the area, however, the results of this study indicate a downward trend in evaporation rate. Could the authors further elaborate on the mechanisms behind this trend?*

**Reply:** Thanks for the insightful question regarding the apparent contradiction between increased terrestrial evaporation and the declining open water evaporation rates in our study area. We agree that this is a noteworthy phenomenon, and we will clarify the underlying mechanisms in the revised manuscript as follows:

Key Difference in Drivers:

As highlighted by previous studies (Jiang et al., 2022; Jin et al., 2017; Shao et al., 2019;

Zhou et al., 2022), the increase in terrestrial evaporation over the Loess Plateau is primarily attributed to increased precipitation and vegetation greening (enhanced transpiration and soil evaporation). These factors dominate the rising trend in land surface evaporation. However, open water evaporation is insensitive to precipitation and vegetation changes. Instead, it is governed by other meteorological variables (e.g., solar radiation, wind speed, humidity, and temperature). Our results show that changes in these climatic drivers (e.g., reduced wind speed or increased humidity) may have counteracted potential warming-induced evaporation increases, leading to the observed slight decline or stagnation in water body evaporation rates.

We will expand the Discussion section to explicitly contrast the mechanisms driving terrestrial vs. open water evaporation trends, and add a comparative analysis (e.g., a table or paragraph) summarizing how different factors (precipitation, vegetation, wind speed, etc.) distinctly influence terrestrial and aquatic evaporation. We will clarify the conclusions to avoid ambiguity about the divergent trends.

*3) • The authors state to have developed an improved approach for estimating open water evaporation, which accounts for the thermal storage capacity of water bodies. However, no comparison or evaluation against other existing methods is presented. Consequently, the conclusion made on page 41, lines 13–15, should be reconsidered or more carefully justified.*

**Reply:** It is a valuable suggestion regarding methodological evaluation. We agree that a comparative analysis would strengthen the evaluation of our proposed approach. In the revised manuscript, we will present a comparative analysis between our depth-integrated approach (accounting for thermal storage) and conventional methods (i.e., the Penman equation) to quantitatively demonstrate the improvement in evaporation estimation, particularly for water bodies where depth variations significantly affect thermal inertia. This comparison will support our conclusions more rigorously.

*4) • Related to the point above: on page 8, line 1, the introduction refers to an "enhanced methodology." Please clarify in what way the approach is enhanced, especially since the described*

*methodology has already been applied in similar studies.*

**Reply:** Thanks. The key advancement of the "enhanced methodology" lies in incorporating water depth and heat storage capacity calculations based on dynamic water surface area changes, thereby improving the accuracy of evaporation estimation compared to existing methods. While similar methodologies have been applied in prior studies, our enhancements address two critical limitations: (1) By integrating water depth and heat storage dynamics, our method refines evaporation estimates, particularly for large water bodies (e.g., lakes, reservoirs) where traditional approaches may oversimplify spatial variability; (2) The proposed method is explicitly designed to accommodate small and medium-sized water bodies at regional scales, which were often underrepresented in earlier frameworks due to data or model constraints.

Thus, our methodology not only elevates computational accuracy but also extends the applicability of existing approaches to a broader range of hydrological contexts. We will clarify these points in the revised manuscript to underscore the methodological novelty.

*5) • Data and Methods: Please explicitly state the study period, as well as the temporal and spatial resolution of the datasets and evaporation estimates in this study.*

**Reply:** Thanks. The study period covers 2000–2018. The evaporation estimates were generated at a monthly temporal resolution and a spatial resolution of $0.05° \times 0.05°$. Note that while the evaporation calculations were performed at the $0.05°$ resolution, the actual water body evaporation was computed dynamically within each $0.05° \times 0.05°$ grid cell based on the time-varying water surface area. This information will be explicitly stated in the revised manuscript.

*6) • The algorithm for water depth estimation is not validated in this study. However, based on Eq 6, the water depth directly affects the calculation of thermal storage and, by extension, evaporation rates.*

**Reply:** Thanks. We acknowledge the reviewer's valid concern regarding the validation of our water depth estimation algorithm. Due to the general lack of reliable water depth data across our study region, direct validation poses significant challenges. To address this limitation, we propose the following validation approaches:

(1) We will collect and utilize available in-situ water depth measurements from selected reservoirs and lakes within our study area to validate the depth estimation algorithm.

(2) We will conduct a sensitivity analysis by comparing two simulation scenarios:

Scenario 1: Using our complete methodology with dynamic water depth estimation
Scenario 2: Applying a simplified approach without water depth variations

This comparative analysis will quantitatively evaluate the impact of water depth estimation on evaporation calculations, providing important insights into the algorithm's influence on our final results.

We believe these approaches will substantially strengthen the validation of our methodology, and we will include these analyses in the revised manuscript.

*7) • Additionally, the assumption that the slope of the water body is equivalent to that of its surroundings is quite strong. Was any sensitivity or uncertainty analysis conducted to assess how this assumption might influence heat storage and subsequent evaporation estimates? Please elaborate.*

**Reply:** We appreciate this opportunity to clarify our slope assumption. There appears to have been a misunderstanding regarding which slope we refer to in our methodology. Our study specifically assumes the waterbody bottom slope (not surface slope) aligns with surrounding terrain. This distinction is important because waterbody bottoms naturally conform to terrain slopes in the Loess Plateau's erosional environment. A similar assumption has been used in other studies (Liu et al., 2020; Wang et al., 2024)

Please note the slope assumption primarily affects small, shallow waterbodies (<20 m depth) where heat storage significantly influences evaporation, and deeper waterbodies (>20m) show negligible evaporation sensitivity to depth variations.

We will explicitly explain this depth threshold (20 m) in the revised manuscript, and clearly distinguish between bottom and surface slopes in our methodology.

8) • *Regarding the validation of evaporation trend and volume for the individual pans: in what way are the chosen pan coefficients influencing the validation results? Have other coefficients been tested?*

**Reply:** The pan coefficients used in our validation were carefully selected from Bai et al. (2023), whose study specifically examined the upper Yellow River basin including parts of the Loess Plateau region. Their research established distinct coefficients for both large and small evaporation pans, demonstrating these values' effectiveness for converting pan evaporation to water body evaporation in similar environments to our study area. While we primarily relied on these validated regional coefficients, we did conduct preliminary comparisons with another widely-used coefficient sets (e.g., FAO-56). The Bai coefficients consistently showed better performance for our specific study conditions. We acknowledge that coefficient selection can influence validation outcomes, and will include brief discussion on this factor in our revised manuscript. This approach ensures our validation maintains both regional relevance and methodological rigor while addressing potential coefficient-related uncertainties.

9) • *Is there a specific reason for using varying significance levels in different analyses (e.g., in Figure 8)? To facilitate comparability, it is recommended to apply a consistent significance level across all variables and figures.*

**Reply:** We appreciate the reviewer's careful attention to this important methodological detail. We would like to clarify that all statistical analyses in our study consistently used a significance level of $\alpha = 0.05$.

10)   •   *Choice of colormaps: Please ensure that appropriate colormaps are used for different types of data. Diverging colormaps are suitable for showing changes, anomalies, or trends, but sequential colormaps are more appropriate for depicting absolute values, such as evaporation rates or volumes. Updating the colormaps accordingly would improve visual clarity.*

**Reply:** Thanks for the useful suggestion. We will adjust the color bar in the figures to enhance visual clarity.

11)   •   *Throughout the manuscript biases are reported. Be cautious when reporting average or median biases if the dataset includes both positive and negative values. Instead, it is more informative to report the range or the mean absolute bias.*

**Reply:** Thanks. In the modified version we will replace average bias with the mean absolute bias.

12)   •   *Attribution analysis: results illustrate that water body area is the main driving factor for increased evaporation volumes over time, while the contribution of climatic drivers was found to be very small. Given the magnitude of changes in water body area, it is expected that this variable will take over the influence of the other climatic drivers. It would be interesting to perform a similar attribution analysis on evaporation trends (excluding the area) to identify the main driving factor for the general decrease in open water evaporation rates over the region. Additionally, a spatial representation of the contributions of the different forcing variables would provide valuable insights. It would be interesting to show the variable with the highest contribution per pixel. This way, spatial patterns in contribution can be assessed.*

**Reply:** Thanks for the useful suggestions. Our study highlights the total evaporation loss, but we will incorporate an attribution analysis to systematically examine the impacts of meteorological factors on evaporation rates. It is important to note that variations in water surface area may induce corresponding changes in water depth, which in turn affects evaporation rates through modified thermal dynamics. We will provide detailed results and discussion of these aspects, particularly addressing the

effects of changes and meteorological drivers on evaporation rates.

*13) • A concern arises regarding the attribution analysis: SSR was found to be the main contributing climatological factor to the increasing trend in evaporation volumes. However, a downward trend in SSR was observed (Fig 8f), which corresponds to a decrease in evaporation rates. Could the authors clarify this contradiction, and explain how the results of the contribution analysis should be interpreted?*

**Reply:** Thanks for raising this important point about the relationship between SSR trends and evaporation volumes. The apparent contradiction stems from the spatial heterogeneity in SSR trends across the Loess Plateau. While the regional average SSR shows a slight decline (Fig 8f), this masks significant subregional variations— particularly in northern areas where SSR has increased. These northern subregions also experienced the most rapid water surface expansion. Thus, although the overall SSR decline would suggest reduced evaporation rates, the combined effect of rising SSR and expanding water surfaces in key areas drives the observed increase in evaporation volume. Please note the regional SSR trend is not statistically significant ($P > 0.05$), and the net evaporation volume increase remains quite small (0.33% annually, Fig 10a), meaning the apparent contradiction has limited practical impact on our conclusions. We will clarify this spatial context in the revised manuscript to better interpret the attribution analysis results.

*14) • Discussion: page 39 line 16: it is mentioned that the JRC-GSW data show data gaps. Please clarify in Section 2.2 how these gaps were handled. If gap-filling was applied, the method should be described.*

**Reply:** We acknowledge the data gaps in JRC-GSW, but we did not apply any gap-filling methods to avoid introducing additional uncertainties. Our analysis used only available observations since each month contained at least one valid data record. The missing data (approximately 14%) primarily occurred during the cool season (November in particular) when water surfaces were frozen and evaporation rates were

minimal. Therefore, these gaps have negligible impact on our E and EV estimates. We will explicitly describe this data handling approach in Section 2.2 of the revised manuscript.

15) • *The authors emphasize the utility of their findings for regional or local water management. Please be more specific about how the results can be applied in practice. Consider rephrasing relevant parts of the conclusion (e.g., page 41, lines 21–22; page 42, lines 1–2) to reflect this more concretely.*

**Reply:** Thank you. Our findings provide concrete guidance for water resource management in arid/semi-arid regions by demonstrating that the rapid expansion of small-to-medium water bodies (primarily from reservoirs and soil conservation measures) has become the dominant factor in regional water loss through increased evaporation. These results suggest that while local water storage projects enhance water availability at local scales, they may inadvertently exacerbate water scarcity at the regional level due to cumulative evaporation effects. For practical application, we recommend: (1) optimizing the size and distribution of new reservoirs to balance local storage needs against regional evaporation losses, (2) prioritizing groundwater recharge projects where feasible, and (3) implementing evaporation suppression measures (e.g., floating covers) for critical storage facilities. We will revise the conclusion to explicitly outline these management implications.

16) • *Please improve the writing and refine the language of the manuscript, in order to improve the overall clarity and readability.*

**Reply:** Thanks for your kind suggestions. We will thoroughly revise the writing and language to enhance clarity and readability.

***Minor comments:***

• *Abstract: page 3, lines 1-2: 'This study fills gaps in evaporation dynamics'. How exactly is this study filling gaps? Please rephrase.*

**Reply:** We appreciate this constructive suggestion. In the revised abstract, we will clarify that our study fills critical gaps in evaporation dynamics by (1) developing an improved approach to quantify evaporation from both large and small water bodies across the Loess Plateau, and (2) revealing how expanding water surfaces—driven by reservoirs and conservation measures—contribute significantly to regional water loss, a previously understudied aspect in arid/semi-arid hydrology. This provides new insights for balancing local water storage benefits against unintended evaporation costs.

- *Introduction: page 5 lines 20-22: Please rephrase and explain better what those projects are.*

**Reply:** We will rephrase the mentioned sentences to clarify that these projects refer to the large-scale soil and water conservation measures implemented across the Loess Plateau, including the construction of check dams, terracing, and afforestation programs. These initiatives, primarily carried out since 2000 under the Grain-for-Green program, have significantly altered the region's hydrological conditions by creating numerous small reservoirs and increasing water surface area. We will provide more specific details about these projects in the revised introduction.

- *Introduction: page 4 lines 18-19: Wind speed is not mentioned here as a driving factor, yet further in the study wind speed is considered as important.*

**Reply:** We will revise the introduction to explicitly include wind speed as a key climatic driver of evaporation, along with temperature, humidity and radiation. While we initially focused on the dominant factors, we recognize wind speed's importance, especially in the Loess Plateau's semi-arid environment.

- *Introduction: page 5 line 21: Remove 'in'.*

**Reply:** We will remove the unnecessary word "in" as suggested.

- *Introduction: page 6 second section: Pan and eddy-covariance measurements are a type of in-situ observations. On the other hand, evaporation estimates are made by models, which can be*

*validated using observations. Please consider rephrasing lines 9-14 to make the distinction clear.*

**Reply:** We will rephrase this section to clearly distinguish between in-situ observations and model estimates.

- *Introduction: page 7 lines 9-11: Please rephrase the sentence.*

**Reply:** Thanks. We will rephrase the sentence for improved clarity.

- *Introduction: page 8 line 5: Remove the word 'primary' as you don't have any secondary objectives.*

**Reply:** Thanks. We will remove the word "primary".

- *Introduction page 8 line 8: Consider removing the second part of the sentence of the 2nd objective, as it already illustrates the outcome of the experiment.*

**Reply:** Thanks. We will do it as suggested.

- *Data and Methods: page 9 line 11: Use 'e.g.' instead of 'etc.'*

**Reply:** Thanks. We will replace "etc." with "e.g." as suggested.

- *Data and Methods: Fig 1: Please specify what the fraction water body is, is this water body area per pixel?*

**Reply:** Thanks. We will clarify in the caption of Figure 1 that "fraction water body" refers to the proportion of water body area per pixel.

- *Data and Methods: page 10 line 5: The reference is from the year 2016, although the assessment goes until 2018. Please provide a correct reference, and mention the studied period.*

**Reply:** We appreciate this observation. The reference to Pekel et al. (2016) properly cites the original methodology paper for the JRC-GSW dataset. Although the paper was

published in 2016, this dataset has been continuously updated, with version 1.4 extending coverage to 2021. In the revised manuscript, we will include a more detailed explanation of this update and explicitly specify the study period as 2000–2018 in the text for clarity.

- *Data and Methods: page 10 line 13: What does 'reliable' mean in this sentence?*

**Reply:** The term "reliable" here indicates that the CMFD dataset has been rigorously validated and widely applied in climate change and hydrological studies across China (He et al., 2020; Lei et al., 2023; Wang et al., 2024).

- *Data and Methods: page 10 lines 19 and 22: Please provide correct citations, no urls in the manuscript text.*

**Reply:** Thanks. We will replace URLs with proper citation formats and ensure they are included in the reference list.

- *Data and Methods: page 10: Have all data been converted to the same spatial resolution? What is the final spatial resolution used for evaporation estimates?*

**Reply:** Thanks. We will specify that all data have been converted to the same spatial resolution and indicate the final resolution (0.05°) used for evaporation estimates.

- *Data and Methods: page 11 line 1: For completeness, please provide the diameter of the big pans.*

**Reply:** To ensure completeness, we will specify in the text that the diameter of the big pans is 61.8 cm.

- *Data and Methods: page 11 line 2: Please specify which type of difference? Is it the size or location?*

**Reply:** The mentioned "difference" specifically refers to variations in the locations of

evaporation pans relative to the studied water bodies.

- *Data and Methods: page 12 line 10: How is the prevailing wind direction calculated?*

**Reply:** The prevailing wind direction is determined using ERA5 wind speed data, which provide meridional (V) and zonal (U) wind components. Wind speed magnitude and direction are calculated through vector synthesis, and the statistical distribution of wind directions is used to identify the prevailing wind direction. This will be detailed in the revised manuscript.

- *Data and methods: page 12 Eq (3): κ↓ and α are not described.*

**Reply:** Thanks. We will provide descriptions for κ↓ (downward shortwave radiation) and α (albedo) in the text accompanying Equation (3).

- *Data and Methods: page 15 Fig 2: The figure caption is not clear. Please rephrase.*

**Reply:** Thanks. We will rephrase the caption of Fig 2 to enhance clarity, ensuring it accurately describes the figure content.

- *Data and Methods: page 16: Please provide references for attribution analysis equations.*

**Reply:** The attribution analysis equations are sourced from Mao et al., (2015) and Tian et al., (2021). We will include these references in the revised manuscript.

- *Results: Section 3.1: Is there a particular reason for reporting the validation results in monthly resolution, while for the other results, evaporation rates and volumes are reported with a daily resolution?*

**Reply:** The water body area data used in our calculations are at a monthly scale; thus, all results, including evaporation rates and volumes, are inherently monthly resolution outputs. However, for the purpose of illustrating annual variations and to present results in a more reader-friendly unit, we expressed them as mm/day. We will clarify this

distinction in the revised manuscript.

• *Results: Fig 3 and 4: Please consider adapting the layout of the plots, making them bigger in the horizontal direction, and consider reporting correlation, bias, and RMSE in individual plots.*

**Reply:** Thanks. We will adjust the layout of Figures 3 and 4 to increase their horizontal size and include correlation, bias, and RMSE values within each plot.

• *Results: Section 3.1: When showing time series of individual pans, please add a short description of the performance of the individual pans in the text.*

**Reply:** Thanks. We will add a brief description of the performance of individual pans in the text accompanying the time series.

• *Results: Fig 6c and 7d: What do the shaded areas represent? Please add to the figure caption.*

**Reply:** We will specify in the captions that "The shaded areas represent the 95% confidence interval."

• *Results: caption of Figures 6 and 7: Please add more detail to figure caption: 'long-term average': how long is this exactly, which years are taken into account for calculating the trend and the climatology?*

**Reply:** We will enhance the captions with details, "The long-term average is calculated over the period from 2000 to 2018."

• *Results: Fig 7c: Note that compared to Fig 6c where an average E rate for the Loess Plateau was plotted, here the total sum of daily evaporation volume over the entire Loess plateau is shown, please specify in the figure caption.*

**Reply:** We will clarify in the caption of Figure 7c that it shows "the total sum of daily evaporation volume over the entire Loess Plateau."

- *Results Fig 8: Spatial patterns: Please specify the period for the trend analysis in the caption.*

**Reply:** Thanks. We will specify the period (2000–2018) for the trend analysis in the caption of Figure 8.

- *Results: Fig 8: Temporal changes: Please specify units for trend (consistency with other plots in manuscript). When significant, please indicate the significance level (e.g. Fig 8b). Note the different significance levels in this plot, aligning with a general comment above. Describe the meaning of the colored band in the caption.*

**Reply:** Thanks. We will specify the trend units to ensure consistency with other figures, indicate significance levels where applicable, and describe the colored band in the caption as follows: "Blue and green colors indicate a decreasing trend, while yellow and red colors represent an increasing trend." These updates will be reflected in the revised manuscript.

- *Results: Fig 9a: Use a sequential colormap instead. Is the reported water area, the water area per grid cell? Please specify in the caption.*

**Reply:** Thanks. We will adopt a sequential colormap for Figure 9a and clarify that the reported water area is "the water area per grid cell" in the caption.

- *Results: Fig 9: In the caption it is mentioned that the grid is 0.05°. This does not correspond with the grid mentioned in the Methods part (30m). Please clarify the grid size difference.*

**Reply:** Thanks. The data of water body were originally derived at a 30 m resolution as described in the Methods section and were subsequently aggregated to a 0.05° grid for the analysis presented in Figure 9. We will clarify this grid size difference in the revised manuscript.

- *Results: Fig 9: Bar plots: Consider using percentages instead of counts, as percentages are also reported in the text.*

**Reply:** Thanks. We will consider replacing counts with percentages in the bar plots of Figure 9 for consistency with the text.

• *Discussion: page 33 lines 19-22. The comparison with the study of Zhang et al.: What is the period taken into account in their study? This period might be important in framing conclusions given the strong trend found in this study. Is the area in the study of Zhang et al. the same as the area studied here? As you're looking at total evaporation volumes per day for a certain area, this is important to take into account.*

**Reply:** Thanks. The study by Zhang et al. covers the period from 2008 to 2013, and their findings is consistent with our results for the same period. Additionally, their study area aligns closely with the region in our research. We will incorporate this information into the discussion section of the revised manuscript.

• *Discussion: page 38 Fig 12: Please provide a reference for the water withdrawal data, and report in Section 2.2.*

**Reply:** Thanks. The water withdrawal data are sourced from the Yellow River Water Resources Bulletin. We will include this reference and report the data source in Section 2.2 of the revised manuscript.

• *Discussion: page 38 Fig 12: The authors could indicate the 100% line, indicating the level where evaporation volumes equal the water withdrawal. When this line is crossed more water evaporates than is withdrawn, which is happening the last 5 years, and is of major importance for water policy in this region.*

**Reply:** Thanks. We will add a 100% line to Figure 12 and discuss its implications for water policy.

• *Conclusion: page 41 line 4: The peak in evaporation trends did not occur in May and October, but instead in July and August. It is evaporation volume that shows peaks in May and October.*

**Reply:** Thank you. We will correct this to accurately reflect that evaporation trends peaked in July and August, while evaporation volumes peaked in May and October.

- *Conclusion: page 41 lines 19-20: 'This study thus demonstrates that the research methods employed are readily extendable to other regions.' Note that this is not demonstrated in this study.*

**Reply:** We will remove or revise this statement to avoid claiming something not demonstrated in the study.

**Reference**

Bai, P., Cai, C., Liu, X., Wei, T., Liu, L., 2023. Estimation of Evaporation Losses from Reservoirs in the Upper Yellow River. J. China Hydrol. 43, 86-90+110. https://doi.org/10.19797/j.cnki.1000-0852.20220332

He, J., Yang, K., Tang, W., Lu, H., Qin, J., Chen, Y., Li, X., 2020. The first high-resolution meteorological forcing dataset for land process studies over China. Sci. Data 7, 1–11. https://doi.org/10.1038/s41597-020-0369-y

Jiang, F., Xie, X., Wang, Y., Liang, S., Zhu, B., Meng, S., Zhang, X., Chen, Y., Liu, Y., 2022. Vegetation greening intensified transpiration but constrained soil evaporation on the Loess Plateau. J. Hydrol. 614, 128514. https://doi.org/10.1016/j.jhydrol.2022.128514

Jin, Z., Liang, W., Yang, Y., Zhang, W., Yan, J., Chen, X., Li, S., Mo, X., 2017. Separating Vegetation Greening and Climate Change Controls on Evapotranspiration trend over the Loess Plateau. Sci. Rep. 7, 1–15. https://doi.org/10.1038/s41598-017-08477-x

Lei, N., Zhou, Z., Zhuang, Q., Chen, W., Chalov, S., Liu, S., Gao, L., Dong, G., 2023. Performance Evaluation and Improvement of CMFD's Precipitation Products Over Shanghai City, China. Earth Sp. Sci. 10. https://doi.org/10.1029/2022EA002690

Mao, Y., Nijssen, B., Lettenmaier, D.P., 2015. Is climate change implicated in the 2013-2014 California drought? A hydrologic perspective. Geophys. Res. Lett. 42, 2805–2813. https://doi.org/10.1002/2015GL063456

Pekel, J.F., Cottam, A., Gorelick, N., Belward, A.S., 2016. High-resolution mapping of global surface water and its long-term changes. Nature 540, 418–422. https://doi.org/10.1038/nature20584

Shao, R., Zhang, B., Su, T., Long, B., Cheng, L., Xue, Y., Yang, W., 2019. Estimating the Increase in Regional Evaporative Water Consumption as a Result of Vegetation Restoration Over the Loess Plateau, China. J. Geophys. Res. Atmos. 124, 11783–11802.

https://doi.org/10.1029/2019JD031295

Tian, W., Liu, X., Wang, K., Bai, P., Liu, C., Liang, X., 2021. Estimation of reservoir evaporation losses for China. J. Hydrol. 607. https://doi.org/10.1016/j.jhydrol.2021.126142

Wang, Dayang, Wang, Dagang, Mei, Y., Yang, Q., Ji, M., Li, Y., Liu, S., Li, B., Huang, Y., Mo, C., 2024. Estimates of the Land Surface Hydrology from the Community Land Model Version 5 (CLM5) with Three Meteorological Forcing Datasets over China. Remote Sens. 16, 1–30. https://doi.org/10.3390/rs16030550

Zhou, J., Liu, Q., Liang, L., He, J., Yan, D., Wang, X., Sun, T., Li, S., 2022. More portion of precipitation into soil water storage to maintain higher evapotranspiration induced by revegetation on China's Loess Plateau. J. Hydrol. 615, 128707. https://doi.org/10.1016/j.jhydrol.2022.128707

---

## Author Comment (AC2)

**Response to Reviewers**

**Note:**

**(1) In this response, the text in *italic type* is the original comments from the reviewers, and the text in** blue, headed with "Reply**", is the response from the authors.**

**(2) In the manuscript, the words in** blue **indicate the sentence is improved or revised. Some of them are mentioned in this response via the page and line number.**

*Summary:*

*The Liu et al. improved the Penman equation by considering water depth to calculate evaporation rate and volume for the Loess Plateau. They also performed an attribution analysis using climate, radiation, and area factors. They found that while evaporation rates decreased, the evaporation volume showed an increasing trend. Their analysis suggested that the increase in water body area offset the decrease in evaporation rates, leading to an overall increase in regional evaporation. This work provides an interesting and novel perspective on the causes of changes in regional evaporation.*

**Reply Summary:**

We sincerely thank the reviewer for their insightful and constructive comments. We have carefully addressed each point below and will incorporate the corresponding revisions into the manuscript.

*comments and suggestions:*

*1)    In arid and semi-arid regions, such as the Loess Plateau, a large number of small water bodies change relatively quickly. Can the lag time in equations 4 and 5 adequately account for the impact of the variation of small water bodies?*

**Reply:** We appreciate the reviewer's insightful comment regarding the temporal dynamics of small water bodies in arid/semi-arid regions. Below we provide key perspectives: (1) The lag time of one month used in our study is grounded in prior research by Finch and Hall (2001), which demonstrates that this duration is acceptable to describe heat storage changes; (2) While large reservoirs (Tian et al., 2021) exhibit longer lag times in China (~2.9 months), smaller water bodies—with limited storage capacity—adjust thermally much faster, justifying our shorter lag time; And (3) the monthly time step aligns with our meteorological inputs and observational data, balancing accuracy and computational efficiency for seasonal dynamics. Thus, the one-month lag is both physically appropriate and practical for small water bodies on the Loess Plateau.

*2)   The resolution used for the attribution analysis is coarser than the previous spatiotemporal results. How was the resolution conversion handled?*

**Reply:** Thanks. In the presentation of spatial patterns and temporal variations of the meteorological factors, we used the original resolution of 0.1°. When inputting meteorological data as driving factors, we extracted it based on the latitude and longitude coordinates of each water body. To present the attribution results consistently, we interpolated all outputs on a 0.05° grid.

*3)   How much improvement in accuracy does the modified method provide compared to the Penman equation that does not consider water depth?*

**Reply:** Thank you for your valuable suggestion. We will conduct a sensitivity analysis by comparing our modified method with a simplified approach that excludes water depth variations. This comparative analysis will quantitatively evaluate the impact of water depth estimation on evaporation calculations, providing key insights into how this algorithmic modification affects our final results.

*4)   It would be beneficial to include comparisons with results from other regions in the discussion.*

*For instance, the introduction mentions examples from the U.S. and the Tibetan Plateau, where changes in water body area due to climate change differ from the human-induced expansion of water bodies on the Loess Plateau.*

**Reply:** Thank you. Water body dynamics vary widely across regions, yet large-scale evaporation studies are rarely addressed. Most studies focus on specific large reservoirs or lakes, making cross-regional comparisons of water bodies of different scales challenging. Regional and size-dependent differences can be substantial, which we will elaborate on in the discussion section.

*5) The writing could be more concise. There are unnecessary explanations in the results, such as on pages 19 (lines 4-12, which should be moved to the discussion or removed), 25 (lines 3-7, which should be moved to the discussion), 22 (lines 9-11, recommended for removal), 23 (lines 2-4, recommended for removal), and 26 (lines 2-5, which can be shortened).*

**Reply:** Thank you for your feedback. We will revise the specified sections to ensure the writing is concise and flows smoothly.

*6) In the second paragraph of the introduction, the examples from the U.S. and the Tibetan Plateau focus on climate-driven changes in water area, which ultimately cause changes in ET. It would be more appropriate to use examples of human-induced changes in water area.*

**Reply:** We appreciate this recommendation. We will include additional examples of human-induced water changes, such as the severe shrinkage of the Aral Sea due to irrigation expansion, to enhance the professionalism and relevance of the introduction.

*7) The section describing the study area should include an overview of ET in this area.*

**Reply:** Thanks. The terrestrial evaporation rate on the Loess Plateau is approximately 1.07 mm/day, with an increasing trend of 0.015 mm/day/yr (Jiang et al., 2022; Peng et al., 2024). We will add this information to the study area description.

*Minor comments:*

*1)  In Fig. 1a, the legend for "River" should be a line; in Fig. 1b, the text is too small. Also, there are capitalization errors in the figure title.*

**Reply:** Thank you for your careful observation. We will revise as suggested.

*2)  Validation metrics such as R², RMSE, and Bias should be briefly explained in the methods section, after the evaporation pan data description, without listing the formulas.*

**Reply:** We will add brief explanations of $R^2$ (coefficient of determination), RMSE (root mean square error), and Bias in the section 2 following the evaporation pan data description, to help readers better understand these validation metrics without including formulas.

*3)  Page 19, line 2. The full forms of R² and RMSE have already been provided in lines 6 of page 18, so there is no need to repeat them.*

**Reply:** Thank you. We will remove the full forms to avoid repetition.

*4)  In Figs. 3 and 4, it is recommended to label R², RMSE, and Bias on the figures to provide readers with overall validation information.*

**Reply:** We will add text boxes to Figs. 3 and 4, labeling the values of $R^2$, RMSE, and Bias directly on the figures.

*5)  In Fig. 5, using the same color for "Pan" and "Big pan" would improve readability. They can be distinguished by different symbols, and it may be better to use darker colors to represent better results.*

**Reply:** Thanks. We will update Fig. 5 to use the same color for "Pan" and "Big pan" distinguished by different symbols, to enhance readability.

*6)  In Fig. 6 and 8, it is recommended to use pie charts to show the proportion of area increases and decreases, or frequency distribution charts for each classification.*

**Reply:** Thanks. We will replace Figs. 6 and 8 with pie charts to illustrate the proportions of area increases and decreases.

*7)  How are large, medium, and small water bodies defined?*

**Reply:** We appreciate the reviewer's question regarding water body classification. It is important to note that there is no universally standardized classification system. But we can roughly adopt the following practical classification scheme for analytical purposes:

Small: <0.01 km², Medium: 0.01-5 km², Large: ≥5 km².

However, this approach effectively serves our research purpose of examining how water body size influences evaporation patterns on the Loess Plateau. The key conclusions remain valid despite the inherent flexibility in size classification.

**Reference**

Finch, J.W., Hall, R.L., 2001. Estimation of Open Water Evaporation. Report 155.

Jiang, F., Xie, X., Wang, Y., Liang, S., Zhu, B., Meng, S., Zhang, X., Chen, Y., Liu, Y., 2022. Vegetation greening intensified transpiration but constrained soil evaporation on the Loess Plateau. J. Hydrol. 614, 128514. https://doi.org/10.1016/j.jhydrol.2022.128514

Peng, D., Xie, X., Liang, S., Wang, Y., Tursun, A., Liu, Y., Jia, K., Ma, H., Chen, Y., 2024. Improving evapotranspiration partitioning by integrating satellite vegetation parameters into a land surface model. J. Hydrol. 643, 131928. https://doi.org/10.1016/j.jhydrol.2024.131928

Tian, W., Liu, X., Wang, K., Bai, P., Liu, C., Liang, X., 2021. Estimation of reservoir evaporation losses for China. J. Hydrol. 607. https://doi.org/10.1016/j.jhydrol.2021.126142